# INTERGROWTH-21st Project international INTER-NDA standards for child development at 2 years of age: an international prospective population-based study

Michelle Fernandes ORCID ,[1,2] José Villar,[2,3] Alan Stein,[4] Eleonora Staines Urias,[2] Cutberto Garza,[5] Cesar G Victora,[6] Fernando C Barros,[7] Enrico Bertino,[8] Manorama Purwar,[9] Maria Carvalho,[10] Francesca Giuliani,[11] Katharina Wulff,[12,13] Amina A Abubakar,[14] Michael Kihara,[15] Leila Cheikh Ismail,[16] Luis Aranzeta,[17] Elaine Albernaz,[18] Naina Kunnawar,[9] Paola Di Nicola,[11] Roseline Ochieng,[10] Tamsin Sandells,[2] Sandy Savini,[2] Sophie Temple,[2] Elizabeth Murray,[4] Eric O Ohuma,[2,19] Michael G Gravett,[20] Ruyan Pang,[21] Yasmine A Jaffer,[22] Julia Alison Noble,[23] Adele Winsey,[2] Ann Lambert,[2] Aris T Papageorghiou,[2,3] Zulfiqar Bhutta,[24] Stephen Kennedy[2,3]

For numbered affiliations see end of article.

**Correspondence to**
Dr Michelle Fernandes;
michelle.fernandes@wrh.ox.ac.uk

## ABSTRACT

**Objectives** To describe the construction of the international INTERGROWTH-21st Neurodevelopment Assessment (INTER-NDA) standards for child development at 2 years by reporting the cognitive, language, motor and behaviour outcomes in optimally healthy and nourished children in the INTERGROWTH-21st Project.

**Design** Population-based cohort study, the INTERGROWTH-21st Project.

**Setting** Brazil, India, Italy, Kenya and the UK.

**Participants** 1181 children prospectively recruited from early fetal life according to the prescriptive WHO approach, and confirmed to be at low risk of adverse perinatal and postnatal outcomes.

**Primary measures** Scaled INTER-NDA domain scores for cognition, language, fine and gross motor skills and behaviour; vision outcomes measured on the Cardiff tests; attentional problems and emotional reactivity measured on the respective subscales of the preschool Child Behaviour Checklist; and the age of acquisition of the WHO gross motor milestones.

**Results** Scaled INTER-NDA domain scores are presented as centiles, which were constructed according to the prescriptive WHO approach and excluded children born preterm and those with significant postnatal/neurological morbidity. For all domains, except negative behaviour, higher scores reflect better outcomes and the threshold for normality was defined as ≥10th centile. For the INTER-NDA's cognitive, fine motor, gross motor, language and positive behaviour domains these are ≥38.5, ≥25.7, ≥51.7, ≥17.8 and ≥51.4, respectively. The threshold for normality for the INTER-NDA's negative behaviour domain is ≤50.0, that is, ≤90th centile. At 22–30 months of age, the cohort overlapped with the WHO motor milestone centiles, showed low postnatal morbidity (<10%), and vision

## Strengths and limitations of this study

► The prescriptive WHO approach for developing biological standards was applied to a population-based sample of healthy and well-nourished children from Brazil, India, Italy, Kenya and the UK to construct the INTERGROWTH-21st Project neurodevelopmental assessment (INTER-NDA) standards for child development.

► Comprehensive health, growth and neurodevelopmental data were prospectively collected, from early pregnancy to 2 years postbirth, providing a unique opportunity to confirm the cohort's health and nutritional status and to control for multiple risk factors associated with suboptimal child development.

► The INTER-NDA is a mixed-methodology, multidimensional, standardised measure of early child development, which can be administered rapidly, by non-specialists in high-income, middle-income and low-income settings.

► The INTER-NDA is a standardised screening assessment and does not provide a clinical diagnosis.

► The age range of the INTER-NDA is 22–30 months and limits its generalisability to other age groups.

outcomes, attentional problems and emotional reactivity scores within the respective normative ranges.

**Conclusions** From this large, healthy and well-nourished, international cohort, we have constructed, using the WHO prescriptive methodology, international INTER-NDA standards for child development at 2 years of age. Standards, rather than references, are recommended for population-level screening and the identification of children at risk of adverse outcomes.

## INTRODUCTION

Approximately 250 million children under the age of 5 worldwide are at risk of not achieving their developmental potential.[1] Effective interventions are available but maximising their benefit at scale depends on identifying those children at greatest need, preferably using standardised methodology.[2]

At present, a multiplicity of methods is used to measure neurodevelopment during early childhood (online supplementary information S1).[3 4] Many of these are administered by specialist staff and were developed using children from either high-income countries or specific low-income and middle-income countries, each drawing their normative sample (often country or region specific) from the respective settings (online supplementary information S1 and S2).[3 4] To our knowledge, none of these tools commonly used to measure neurodevelopment in early childhood, were based on children monitored from fetal life and have adopted the prescriptive approach recommended by the WHO for the development of international biological standards during the construction of their norms.[5] Instead, references have been commonly used to assess the overall achievement of developmental skills and track progress over time in both, groups of children and individuals. However, while references describe *how* children, in a specific setting and time, have attained certain milestones of interest, they do not describe how children, in all settings, *should* develop. The importance of this fundamental difference between references and standards was elegantly highlighted, in the context of skeletal growth in young children, by WHO Multicentre Growth Reference Study (MGRS), which resulted in the construction of prescriptive international standards for monitoring child growth.[6] These WHO standards, describing optimal growth from early pregnancy to 5 years of age,[7] are now widely employed in clinical practice and used to make comparisons across disparate populations.

The construction of international, prescriptive standards describing optimal neurodevelopment during early childhood is challenging not least because of the technical and logistical difficulties of implementing comprehensive early child developmental assessments across large international populations. To construct international *standards* of child development, in accordance with WHO's prescriptive methodology,[5] four fundamental methodological principles must be fulfilled: (1) the normative sample should be selected using a 'prescriptive' approach, which includes consideration of key factors known to be associated with poor developmental outcomes during early childhood (online supplementary information S2); (2) the conceptual framework *must* be population-based and international; (3) rigorous data management, standardisation and quality control procedures must be included and (4) measurements must be complemented by independent assessments of specific functional and developmental domains (eg, tests of vision) to confirm the prescriptive characteristics of the sample. This rigorous approach is important because the inclusion of inadequately nourished children, or those with mild neurodevelopmental disturbances (NDDs), in normative samples, can affect resultant thresholds. Moreover, the identification of children at risk of (even mild) NDDs is essential because there is evidence to show that very small developmental differences between individuals during early childhood can result in marked discrepancies in mental and physical health, educational attainment, and social and economic outcomes during later life.[8 9]

The INTERGROWTH-21st Project aimed to adopt this WHO prescriptive approach in constructing international standards for child development measured on a standardised, comprehensive assessment tool—the INTERGROWTH-21st Neurodevelopment Assessment (INTER-NDA)—at 2 years of age. Despite this circumscribed age range, by leveraging on the INTERGROWTH-21st Project's international cohort of mothers and children, recruited specifically to be optimally healthy and well-nourished throughout the duration of pregnancy and confirmed, during the infant follow-up component of the project, to be at low risk of adverse birth, health and growth outcomes at birth, 1 and 2 years of age, we were able to adopt the prescriptive approach and methods recommended by the WHO MGRS in the construction of the INTER-NDA *standards* of child development at 2 years of age. In the present study, we analysed cognitive, language, motor and behaviour outcomes at 2 years of age, measured on the INTER-NDA, for healthy and well-nourished children from the INTERGROWTH-21st Project study sites in Brazil, India, Italy, Kenya and the UK. We compared the vision, gross motor, attentional problems and emotional reactivity profiles, as well as growth and health outcomes, in these children to the corresponding norms for these independent measures.

## METHODS

### Study design and population

The INTERGROWTH-21st Project was a multicentre, population-based study conducted between 2009 and 2016, in eight delimited geographical areas worldwide: the cities of Pelotas, Brazil; Turin, Italy; Muscat, Oman; Oxford, UK; Seattle, USA; Shunyi County, a suburban district of the Beijing municipality, China; the central area of the city of Nagpur, Maharashtra, India and the Parklands suburb of Nairobi, Kenya. A geographical area was a complete city, or county, or part of a city with clear political or geographical limits, located at an altitude <1600 m, with low-risk health indicators for perinatal morbidity and mortality, in which women receiving antenatal care had plans to give birth within the area, that had to be free or have low levels of major, known, non-microbiological contamination.[10] The primary aim of the INTERGROWTH-21st Project was to study growth, health and development from early fetal life to 2 years of age in low-risk populations of mothers and children with

optimal health and nutrition so as to produce prescriptive standards of fetal growth, newborn size and early child neurodevelopment to complement the existing WHO Child Growth Standards.

The INTERGROWTH-21st Project recruited pregnant women from the aforementioned populations, who met the individual entry criteria of health, nutrition, education and socioeconomic position (online supplementary information S3).[10] Standardised clinical care and neonatal feeding practices were implemented based on project protocols. The newborn cohort was followed up at birth, 1 and 2 years of age and evaluated for growth, nutrition, health and the WHO gross motor milestones, using standardised methodology and rigorous quality control processes.[11] They constitute the Infant Follow-up Study (IFS) of the INTERGROWTH-21st Project. The baseline characteristics of the full cohort and follow-up methodology have been published elsewhere.[11] The project protocols are available at www.intergrowth21.org.uk.

## Data collection and evaluation methods

All eligible children in five of the eight INTER-GROWTH-21st Project study sites (the cities of Pelotas (Brazil); Turin (Italy); Oxford (UK); Nagpur (India) and the Parklands suburb of Nairobi (Kenya)), who had contributed data towards the construction of the international Fetal Growth and Newborn Growth Standards,[12 13] were invited to attend a comprehensive neurodevelopmental evaluation at the time of their second birthday. This age was selected as it was found to be the earliest at which: (1) neurodevelopment is not confounded by transient neurological syndromes of prematurity and (2) conventionally used developmental instruments, such as the Bayley Scales of Infant Development (BSID), have been found to possess an acceptable level of medium and long-term predictive validity.[14] The sites in China, Oman and the USA did not participate because of logistical and administrative reasons, delays in the start of the study and/or staff availability, all unrelated to the IFS' main hypotheses (a comparison in the demographics, and health and growth outcomes between these sites has already been published).[11]

The evaluation consisted of (in order of administration): an assessment of vision (the Cardiff tests) an assessment of cognition, motor skills, language skills and behaviour (the INTER-NDA); caregiver reports of attentional problems and emotional reactivity (the corresponding subscales of the preschool Child Behaviour Checklist; CBCL); measurement of cortical auditory processing (to a novelty odd-ball paradigm on a wireless, gel-free electroencephalography system); measurement of infant sleep (using actigraphy) and an assessment of gross motor milestones (based on the WHO's checklist). Despite measuring cortical auditory processing and sleep in our cohort, a description of the methods and results relating to these technically complex outcomes are beyond the scope of this paper. Moreover, as normative values for cortical auditory evoked response potentials

and actigraphy data do not exist for children aged 2 years, the added value of these measures in confirming the healthy and well-nourished status of the cohort is uncertain. Information on the child's health and nutritional status, and anthropometric measurements (weight, length and head circumference), were also collected, at the 2 year visit, according to the INTERGROWTH-21st Project protocols.

A specially designed training programme for the neurodevelopmental evaluation was implemented at all sites between 2012 and 2013.[15] Staff administering the assessments were aware of the project's general principles but not the specific hypotheses being tested. They were also unaware of individual children's scores from their own and other study sites.

## Primary outcome measure: the INTER-NDA

The INTER-NDA is a comprehensive, rapid assessment of cognition, (fine and gross) motor skills, language and (positive and negative) behaviour for children aged 22–30 months (online supplementary information S4).[15] Its 37 items are administered in approximately 15 min using a combination of psychometric techniques (direct administration, concurrent observation and caregiver reports) to minimise risks of reporter and recall bias commonly encountered in caregiver interviews[3] while acknowledging that children might perform differently in artificial testing environments than in familiar settings. Children's performance on the INTER-NDA is scored across a spectrum of abilities, rather than on a predefined checklist and, therefore, affords a wider description of a child's faculties.[15] It has demonstrated strong agreement with the BSID, third edition (BSID-III) (interclass correlation coefficients 0·75–0·88, p<0·001 for all domains with little to no bias on Bland Altman analysis); satisfactory internal consistency (Cronbach's alpha 0.56–0.81) and good unidimensionality across subscales (Comparative Fit Index=0.90; Tucker-Lewis Index=0 .94)[16] and good levels of inter-rater (k=0·70; 95% CI 0·47 to 0·88) and test–retest reliability (k=0·79; 95% CI 0·48 to 0·96).[15]

The INTER-NDA is designed for use across socioeconomic groups and populations. Its operation manual, standardisation protocol and forms are freely available at www.intergrowth21.org.uk. The kit consists of common household items encountered across the world. In all study sites, the INTER-NDA was translated into the local languages of the sites (Brazil: Brazilian Portuguese, India: Marathi; Italy: Italian; Kenya: Kiswahili), using the WHO Mental Health Initiative translation guidelines,[17] which included processes of cultural customisation, translation and back translation.

## Other outcome measures of neurodevelopment

To confirm the developmental normality of our cohort, we assessed specific functional and developmental outcomes of relevance by including three measurements independent of the INTER-NDA: (1) visual acuity and contrast sensitivity, measured on the Cardiff Tests[18]; (2)

attention problems and emotional reactivity measured on the respective subscales of the preschool CBCL[19] and (3) the age of achievement of six gross motor milestones measured on the WHO's checklist.[20]

The Cardiff Tests are validated and reliable measures of binocular vision in children that are not influenced by coexisting disturbances in language or cognition, and are independent of cultural biases. Their norms have been applied for clinical purposes.[18] The operational manual for their use in the INTERGROWTH-21st Project is available at https://www.intergrowth21.org.uk. Their administration takes 5 min. Visual acuity and contrast sensitivity are measured in quick succession and taken together are a more robust measure of the integrity and functioning of the entire visual pathway than either test alone.[21]

The preschool version of the CBCL is a parent-rated questionnaire used worldwide as a diagnostic screen for behavioural and emotional problems in young children (https://aseba.org/translations/).[19] In the IFS of the INTERGROWTH-21st Project, mothers completed questions relating to the attentional problems and emotional reactivity CBCL scales.

The WHO Gross Motor milestones checklist consists of the normative windows of achievement for six gross motor milestones, developed from the WHO MGRS cohort between 4 and 24 months of age.[20] In the INTERGROWTH-21st Project, parents were asked to report the age when they first observed or 'never observed' the milestones. The same information was collected from parents at the 1-year and 2-year follow-up visits to evaluate the consistency of the reported dates.[11]

### Data management and statistics

The INTERGROWTH-21st Project neurodevelopmental evaluation was supported by an electronic, tablet-based data collection and management system (the NeuroApp).[15] This contained the INTER-NDA and vision scoring forms, operation manuals, visual cues and integrated data quality checks to facilitate rapid collection of high-quality data and to ensure their secure upload to the project's centralised and site-based data servers on which rigorous monthly checks were performed.[22]

For the INTER-NDA, two standardisation evaluations were carried out, in accordance with guidelines published in the World Bank's Toolkit for Examining Early Child Development,[3] to assess the ability of assessors to score and administer the INTER-NDA. During the first evaluation, assessors scored children's skills on the INTER-NDA from video recordings of four assessments performed by an expert assessor. Inter-rater and test–retest reliability were compared between assessors. At the second evaluation, an expert observed assessors performing three assessments each, and rated each assessor for their ability to administer the INTER-NDA correctly on a standardised protocol adherence checklist (online supplementary information S5). Protocol adherence scores were compared between assessors. The results of these evaluations are presented in online supplementary information S6.

The sample size considerations for this report have been previously published and depended on pragmatic considerations.[23] In summary, as the present report is the 2-year follow-up of the initial FGLS cohort of pregnant women, the total number of eligible children assessed at 2 years of age was therefore fixed. The initial sample size estimations (approximately 500 fetuses per site) focused on the precision and accuracy of the extreme centiles of the complete population, that is, the 3rd or 97th centile because they correspond closely to ±2 SD, and they are the recommended cut-offs of the WHO Child Growth Standards, which are used internationally to evaluate children of this age; however, in the present study, such estimations do not apply because of the different nature of the hypothesis.[23] In this component of the study, neurodevelopment was evaluated in an average of 261 children per site (1307 children total) at 2 years of age. This sample size was considered adequate to explore the predicted small site-specific differences. Post hoc power calculations showed that the study was sufficiently powered to observe small differences among study sites (calculations for INTER-NDA domains with power >0.99) and small effect sizes for the between-group variances.[23] For example, for a between-group variance of 10% of the total variance and a two-tail alpha of 0.05, the power is 0.84.

Summary statistics were calculated for birth, neonatal and postnatal characteristics of children completing the neurodevelopmental evaluation and compared with those lost to follow-up. These characteristics include most factors associated with poor child neurodevelopmental outcomes during the first 2 years of life (online supplementary information S2). The analytical and statistical strategy for the construction of the INTER-NDA centiles is presented in figure 1. For all analyses, Stata V.15 software was used (StataCorp).

Data from the participating sites were pooled, following the strategy recommended by WHO.[7] We have previously reported striking similarities in the distribution of the INTER-NDA domains among children from the five sites.[23] In summary, similar to the patterns observed in linear growth from fetal life to childhood, the variability in INTER-NDA scores between sites is far less (for most domains <10%) than the total variability between individuals within a study site, justifying pooling the data to construct international standards.[23 24]

Raw mean INTER-NDA domain scores (online supplementary information S7[16]) were calculated and their distributions explored. These showed important skewness and (particularly) kurtosis. As 30 INTER-NDA items were scored on a five-point scale, and six items were scored on a three-point scale, raw domain scores were converted to standardised scaled scores (online supplementary information S8).

To explore the low-risk profile of the cohort, centiles for visual acuity (measured in logMAR) and contrast sensitivity (measured in contrast per cent) were determined and compared with the Cardiff Tests' established norms.[18] Attention problem and emotional reactivity

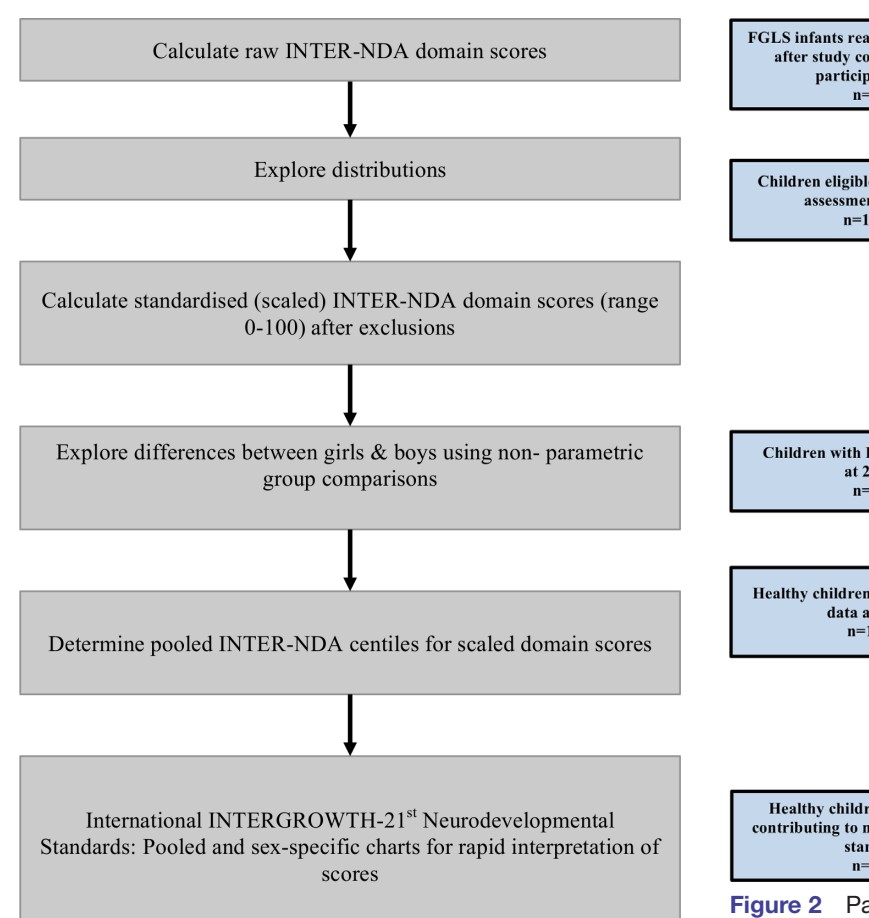

**Figure 1** Analytical and statistical strategy for the construction of the International INTERGROWTH-21st Project INTER-NDA standards. INTER-NDA, INTERGROWTH-21st Neurodevelopment Assessment.

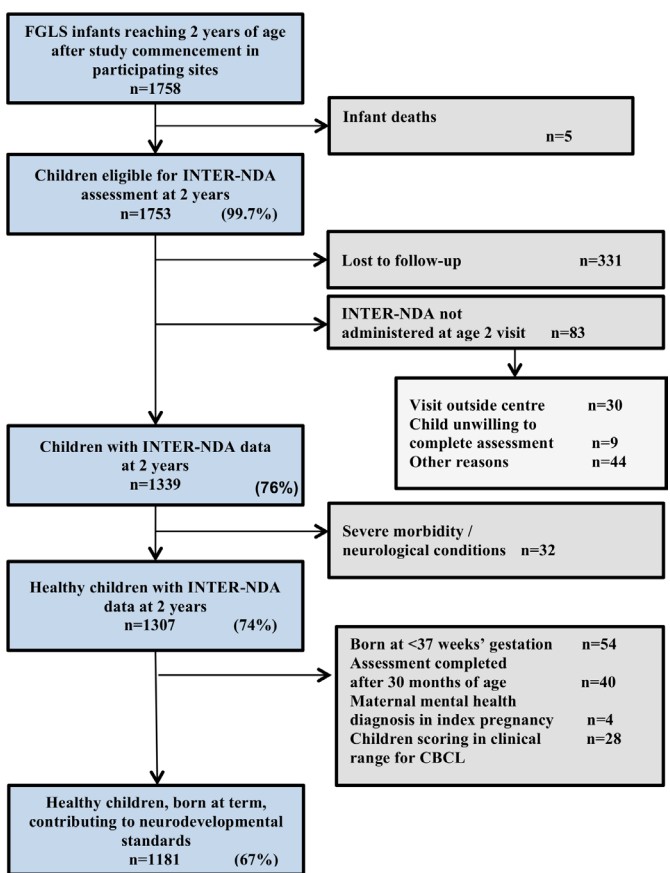

**Figure 2** Participant flow for the INTERGROWTH-21st Project Infant Follow-up Study cohort at 2 years of age. CBCL, Child Behaviour Checklist; FGLS, Fetal Growth Longitudinal Study; INTER-NDA, INTERGROWTH-21st Neurodevelopment Assessment.

subscale scores were calculated using ASEBA-web software, and compared with the CBCL's norms for group 1 societies.[19] The proportion of children within the WHO motor development windows of achievement was estimated as previously described.[11]

In addition, after other exclusions (figure 2), 28 of 1209 eligible children scored above the CBCL's 97th percentile threshold for clinical problems on the attentional problems and emotional reactivity CBCL subscales. We compared INTER-NDA centiles including and excluding this group (online supplementary information S9). As the INTER-NDA centiles were marginally lower on some domains when this group was included, we decided to exclude these children from the normative sample INTER-NDA sample in the construction of the INTER-NDA domain standards.

As no transformation was identified that suited all INTER-NDA domains, the Harrell-Davis distribution-free estimator was used to estimate pooled centiles from the standardised scaled scores.[25] This estimator weights the order statistics by the difference between two incomplete beta functions. INTER-NDA scaled domain scores were compared between boys and girls using the Wilcoxon rank-sum test.

### Patient and public involvement
Patients were not involved in developing plans for the design of the study. Parents showed support for the study through high and sustained follow-up rates in all study sites. The INTERGROWTH-21st Project maintains contact with parents in the cohort through newsletters, webinars and blogs on its website, https://intergrowth21.tghn.org/ and through Twitter (@intergrowth21st).

## RESULTS
### The INTERGROWTH-21st Project IFS Study: INTER-NDA normative cohort characteristics and overall health and nutrition at 2 years of age
#### Population
Of the 1758 eligible children enrolled in the five participating sites, 1339 (76%) were assessed at 2 years of age (figure 2). After exclusions (including 54 children (3.1%) who were born at $<37^{+0}$ weeks' gestation and 28 children who scored at the threshold for clinical problems on the attentional problems and/or emotional reactivity subscales of the CBCL), data from 1181 healthy children (67% of those eligible) were pooled to construct the international INTERGROWTH-21st Neurodevelopmental Standards. The study sites in Brazil, India,

**Table 1** Prenatal, perinatal and neonatal characteristics of children who completed the INTER-NDA in the INTERGROWTH-21st Project compared with those lost to follow-up

| Prenatal, perinatal and neonatal characteristics | Children contributing to INTERGROWTH-21st international INTER-NDA standards (n=1209) Mean (SD) or number (%) | Children lost to follow-up (n=331) Mean (SD) or number (%) |
|---|---|---|
| Maternal age at recruitment, years | 28.4 (3.8) | 27.4 (4.3) |
| Maternal body-mass-index, kg/m$^2$ | 23.2 (3.0) | 23.6 (2.8) |
| Multiple gestation | n=0 (0.0%) | n=0 (0.0%) |
| Chronic maternal illness | n=96 (8.1%) | n=26 (7.9) |
| Maternal infections (including HIV, rubella, syphilis, hepatitis B, cytomegalovirus, toxoplasmosis, tuberculosis and malaria) | n=0 (0.0%) | n=1 (0.3) |
| Maternal haemoglobin (g/L) | 124.0 (10.0) | 124.0 (10.0) |
| Maternal malignancy | n=0 (0.0%) | n=0 (0.0%) |
| Maternal substance abuse (including alcohol) and smoking | n=0 (0.0%) | n=0 (0.0%) |
| Maternal use of teratogenic drugs during pregnancy | n=628 (53.2) | n=222 (67.1%) |
| Maternal prenatal anxiety and depression/mental stress | n=0 (0.0%) | n=0 (0.0%) |
| Maternal preeclampsia and eclampsia | n=10 (0.9%) | n=4 (1.2%) |
| Placental structural anomalies | n=0 (0.0%) | n=0 (0.0%) |
| Foetal growth restriction | n=67 (5.7%) | n=14 (4.2) |
| Gestational age at delivery, weeks* | 39.6 (1.2) | 39.3 (1.5) |
| Birth weight, kg* | 3.2 (0.4) | 3.2 (0.5) |
| Birth length, cm* | 49.2 (1.8) | 49.0 (2.1) |
| Head circumference at birth, cm* | 34.0 (1.2) | 34.0 (1.3) |
| Apgar at five min* | 9.5 (0.6) | 9.6 (0.7) |
| Age at hospital discharge, days† | 3.0 (2.0 to 4.0) | 2.0 (1.0 to 3.0) |
| Boys* | n=564 (47.8) | n=160 (48.3) |
| Hyperbilirubinaemia* | n=49 (4.1) | n=18 (5.5) |
| Respiratory distress syndrome* | n=16 (1.4) | n=7 (2.1) |
| Transient tachypnoea of the newborn* | n=11 (0.9) | n=12 (3.6) |
| Exclusive breastfeeding at hospital discharge* | n=1097 (93.0) | n=300 (90.9) |

Data are mean (SD) or number (%) unless otherwise specified.
Missing data below 2% for all variables.
*Mean (SD).
†Median (IQR).
INTER-NDA, INTERGROWTH-21st Neurodevelopment Assessment.

Italy, Kenya and the UK respectively contributed 147 (12.2%), 305 (25.2%), 296 (24.5%), 301 (24.9%) and 160 (13.2%) children to the normative INTER-NDA cohort. A detailed description of the prenatal, birth, post-natal morbidity, growth and nutritional characteristics of the cohort, during the first 2 years of life, has previously been published and is presented in tables 1 and 2.[11] The comparison in sociodemographic, birth, health and growth characteristics between the five sites that contributed to the normative INTER-NDA cohort, and the three sites that did not, has also been previously published— no significant differences were observed between the two groups.[11]

The mean (±SD) age of both girls and boys at assessment was 24.8 (±1.6) months. Eighty-nine per cent of the neurodevelopmental measures were obtained between 22 and 24 months of age, and 99.9% between 22 and 30 months. The baseline prenatal, perinatal and neonatal characteristics were very similar across the five sites,[23] and with those children lost to follow-up (table 1).

### Health, growth and nutritional outcomes from birth to 2 years

The cohort's mean gestational age and weight at birth were 39.6 (±1.2) weeks and 3.2 (±0.4) kg, respectively.[11] The mean birth length and head circumference were 49.2 (±1.8) cm and 34.0 (+1.3) cm, respectively. Mean age

**Table 2** Postnatal morbidity between 1 and 2 years of age of children contributing to INTERGROWTH-21st international INTER-NDA standards

| Morbidity between 1 and 2 years of life | Children contributing to INTERGROWTH-21st international INTER-NDA standards (n=1209) |
|---|---|
| Hospitalised at least once | 113 (9.4) |
| Total no of days hospitalised* | 2 (1–3) |
| Any prescription provided by a healthcare practitioner | 712 (59.1) |
| Antibiotics (≥3 regimens) | 142 (11.8) |
| Iron/folic acid/vitamin $B_{12}$/other vitamins | 194 (16.1) |
| Up to date with local vaccination policies | 1136 (94.4) |
| Otitis media/pneumonia/bronchiolitis | 88 (7.3) |
| Parasitosis/diarrhoea/vomiting | 43 (3.6) |
| Exanthema/skin disease | 150 (12.5) |
| Urinary tract infection/pyelonephritis | 5 (0.4) |
| Fever ≥3 days (≥3 episodes) | 134 (11.1) |
| Other infections requiring antibiotics | 40 (3.3) |
| Asthma | 13 (1.1) |
| Gastro-oesophageal reflux | 3 (0.2) |
| Cow's milk protein allergy | 8 (0.7) |
| Food allergies | 13 (1.1) |
| Injury or trauma | 27 (2.2) |
| Any condition requiring surgery | 9 (0.7) |

Data are number (%) unless otherwise specified.
Missing data below 2% for all variables.
*Median (IQR).
INTER-NDA, INTERGROWTH-21st Neurodevelopment Assessment.

at discharge from hospital, postbirth, was 3 (2-4) days. At hospital discharge, 89% of the cohort was exclusively breast milk fed.[11] Exclusive breastfeeding was stopped at a median of 5 months (IQR, 3–6 months) and (any) breast feeding stopped entirely at a median of 12 months (IQR, 6–18 months). Detailed information on the nutritional status of the cohort has been previously published.[11]

The overall postnatal morbidity of the cohort was low (table 2): 9.4% of infants were hospitalised during the second year of life with a median hospital stay of 2 days (IQR 1–3 days). The most frequently morbidities reported in outpatient clinics were exanthema/skin diseases, ≥3 episodes of fever lasting ≥3 days, and otitis media/lower-tract respiratory infections.[11 23] At 2 years of age, 92%, 90% and 91% of the cohort's length, weight and head circumference measures respectively were within the 3rd and 97th centiles of the WHO Child Growth Standards.[11]

### Developmental profile of the cohort on independent measures of vision and development at 2 years

The visual acuity and contrast sensitivity centiles for our cohort are presented in table 3. The cohort's 50th centile values for visual acuity and contrast sensitivity were 0.20 logMAR and 1.5%, respectively. Both are within the Cardiff tests' normative values for binocular visual acuity in children aged 24–30 months.[18] The visual acuity and contrast sensitivity values were identical for boys and girls across all centiles (table 3) suggesting no biological variability in these outcomes between sexes.

The cohort's attentional problems and emotional reactivity scores at the 50th centile corresponded to CBCL T-scores of 53 and 50, respectively, that is, the 62nd and <50th CBCL centiles. These values are below the CBCL's 93rd centile threshold for 'borderline clinical problems'.[19] For these CBCL subscales, 28 (2.1%) FGLS children scored above the CBCL's cut-off for clinical problems (>97th centile). These children were excluded from the INTER-NDA normative sample.

At 2 years of age, the cohort overlapped almost perfectly with the WHO motor milestones at the 50th, 3rd and 97th centiles of the range for healthy term infants.[11] For length and head circumference, the mean+SD z-score was 0.0±1.1 for both measures, and the respective medians were at the 49th and 50th percentiles of the WHO Child Growth Standards.[11] For weight, the mean±SD z-score was 0.2±1.1, and median was at the 58th percentile. A detailed description of these characteristics are presented elsewhere.[11]

**Table 3** Visual acuity and contrast sensitivity centiles, measured on the Cardiff tests, in the normative sample of the International INTER-NDA standards

| | Pooled centiles (n=1209) | Girls (n=628) | Boys (n=581) | P value |
|---|---|---|---|---|
| **Visual acuity (logMAR)** | | | | |
| c10 | 0.3 | 0.3 | 0.3 | |
| c25 | 0.2 | 0.2 | 0.2 | |
| c50 | 0.2 | 0.2 | 0.2 | |
| c75 | 0.1 | 0.1 | 0.1 | |
| c90 | 0.1 | 0.1 | 0.1 | |
| Median (IQR) | 0.2 (0.1–0.2) | 0.2 (0.1–0.2) | 0.2 (0.1–0.2) | 0.463 |
| **Contrast sensitivity (%)** | | | | |
| c10 | 2.0 | 2.0 | 2.0 | |
| c25 | 1.8 | 1.6 | 1.9 | |
| c50 | 1.5 | 1.5 | 1.5 | |
| c75 | 1.0 | 1.0 | 1.0 | |
| c90 | 1.0 | 1.0 | 1.0 | |
| Median (IQR) | 1.5 (1.0–2.0) | 1.5 (1.0–1.5) | 1.5 (1.0–2.0) | 0.303 |

*P value from Wilcoxon rank-sum test.
INTER-NDA, INTERGROWTH-21st Neurodevelopment Assessment.

## International standards for the cognitive, motor, language and behaviour domains of the INTER-NDA

The 3rd, 5th, 10th, 50th, 90th, 95th and 97th centiles for the INTER-NDA standardised (scaled) scores for cognition, language, motor and behaviour domains for healthy, well-nourished 2-year-old children are presented, in table 4, for the pooled cohort. For all INTER-NDA domains, except negative behaviour, higher scores reflect better outcomes and the threshold for normality was defined as ≥10th centile. For negative behaviour, where lower scores reflect better outcomes, the threshold for normality was defined as ≤90th centile. The thresholds of normality for the INTER-NDA's cognitive, fine motor, gross motor, language and positive behaviour domains are ≥38.5, ≥25.7, ≥51.7, ≥17.8 and ≥51.4, respectively. The threshold for normality for the INTER-NDA's negative behaviour domain is ≤50.0. To facilitate the easy and rapid implementation of these standards, in clinical, community and research settings, for the identification of children scoring ≤10th and ≤3rd centile on the INTER-NDA (≥90th and ≥97th centiles for negative behaviour) who would benefit from urgent and routine further assessment and/or specialist referral, respectively, we have developed a neurodevelopmental chart that can be printed or downloaded (figure 3).

INTER-NDA domain scores were similar between the cohort's male and female children (online supplementary information S10). There was a trend towards higher cognitive and language scores among girls, and higher negative behaviour scores among boys (online supplementary information S10); however, the clinical and developmental implications of these differences are unclear.

**Table 4** The INTERGROWTH-21st Project international INTER-NDA standards for child development at 2 years of age

| INTER-NDA domain | Pooled centiles (n=1181) | | | | | | |
|---|---|---|---|---|---|---|---|
| | c3 | c10 | c25 | c50 | c75 | c90 | c97 |
| Cognitive* | 27.4 | 38.5 | 62.2 | 79.5 | 88.8 | 92.6 | 99.6 |
| Fine motor* | 17.5 | 25.7 | 74.2 | 91.4 | 100.0 | 100.0 | 100.0 |
| Gross motor* | 31.1 | 51.7 | 66.7 | 81.6 | 100.0 | 100.0 | 100.0 |
| Language* | 12.1 | 17.8 | 45.7 | 71.7 | 88.5 | 95.1 | 100.0 |
| Positive behaviour* | 37.8 | 51.4 | 70.0 | 90.0 | 100.0 | 100.0 | 100.0 |
| Negative behaviour† | 0.0 | 0.0 | 0.0 | 25.0 | 25.0 | 50.0 | 76.5 |

*For these domains, higher scores reflect better outcomes.
†For negative behaviour, lower scores reflect better outcomes.
INTER-NDA, INTERGROWTH-21st Neurodevelopment Assessment.

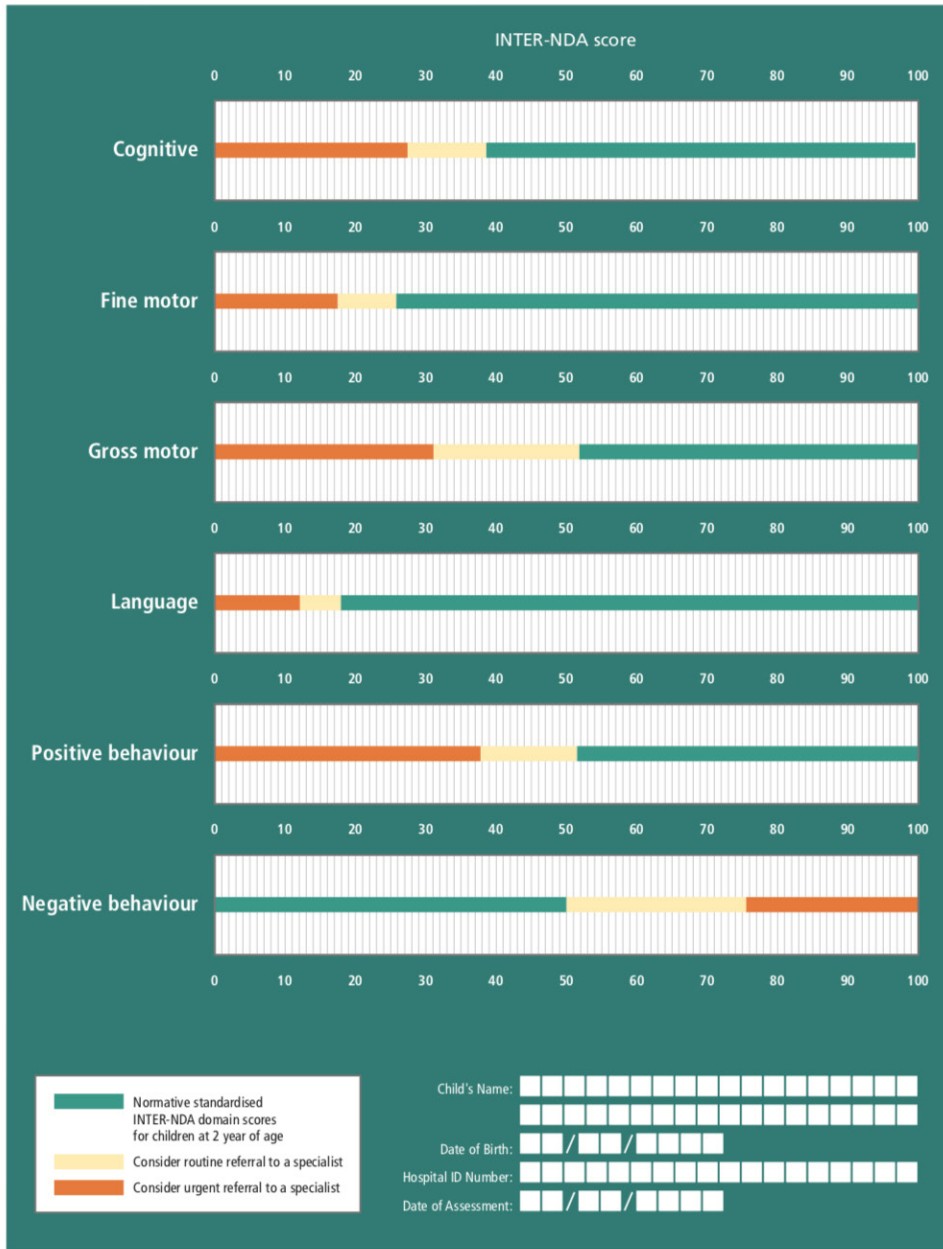

**Figure 3** The INTERGROWTH-21st Project international INTER-NDA standards for child development at 2 years of age. INTER-NDA 3rd to 97th centile ranges for 2-year-old children are presented. These are based on scaled INTER-NDA standardised domain scores. Scores falling in the yellow zone correspond to scores between the 10th and 3rd centiles; scores in the orange zone correspond to scores <3rd centile. Clinical judgement should determine whether further developmental assessment is warranted for children with scores in the yellow and orange zones, and the urgency of such referrals. INTER-NDA, INTERGROWTH-21st Neurodevelopment Assessment.

## DISCUSSION

From this international, population-based cohort of optimally healthy and nourished children from Brazil, India, Italy, Kenya and the UK (online supplementary information S11 and S12); monitored from early pregnancy to 2 years of age, we have constructed international prescriptive standards for cognitive, language, motor and behavioural outcomes in 2-year-old children measured on a rapid, comprehensive assessment—the INTER-NDA. These centiles were constructed after excluding children born at <37 weeks gestation; those with significant/neurological morbidity, those whose mothers were known to have a mental health diagnoses during pregnancy and those who scored above the threshold for

clinical attentional and emotional reactivity problems on the CBCL. We have confirmed the prenatal, perinatal, neonatal and postnatal healthy and well-nourished status of the normative INTER-NDA cohort using multiple measures during pregnancy, birth, 1 and 2 years of age; and have confirmed, at 2 years of age, its low-risk profile or adverse neurodevelopmental outcomes.[11] The threshold of normality for the INTER-NDA's cognitive, fine motor, gross motor, language and positive behaviour domains is ≥38.5, ≥25.7, ≥51.7, ≥17.8 and ≥51.4, respectively. The threshold for normality for the INTER-NDA's negative behaviour domain is ≤50.0. These centiles represent, to our knowledge, the first endeavour to construct standards for child developmental outcomes in this age group using the WHO prescriptive methodology and an international sampling frame. To facilitate the easy and rapid implementation of these standards in clinical, community and research settings for the identification of children at risk who would benefit from routine and urgent further assessment and specialist referral, respectively, we have developed a neurodevelopmental chart that can be printed or downloaded (figure 3).

## Strengths and limitations of this study

The strengths of the IFS Study of the INTERGROWTH-21st Project include the population-based cohort design; the use of the WHO recommended 'prescriptive' approach; the international sampling frame; the inclusion of rigorous data management, standardisation and quality control procedures and the incorporation of independent measurements of specific functional and developmental domains (vision, attentional problems, emotional reactivity and age of acquisition of key gross motor milestones) to confirm the satisfactory growth, health and development of our cohort was confirmed prior to the construction of these standards.[11] In addition, we used the INTER-NDA as the developmental measure of choice to construct these standards (table 5). In online supplementary information S1 and S2, we present an overview of the normative samples and thresholds for NDDs of ten instruments commonly used to measure neurodevelopmental outcomes in 2-year-old children. Of these, two tools (the Guide for Monitoring Child Development, GMCD[26 27] and the Caregiver-reported Early Developmental Instruments, CREDI[28]) fulfil some of the WHO-based methodological criteria for the construction of child developmental standards (GMCD: criteria 1, 3 and 4; and CREDI: criteria 2 and 4). The INTER-NDA fulfils 24 of the 26 criteria. Although a multidimensional assessment is easy to implement and was designed for use across population groups in high-income, middle-income and low-income settings.[15] Despite an administration time of 15 min, it has demonstrated good to acceptable agreement with the BSID-III,[16] and can be administered reliably, in the field, by trained non-specialists (online supplementary information S6).

The main limitation of our study was that the INTER-NDA is restricted to the 2-year age group. We selected 22–30 months as the time point for the key developmental assessment of the entire study because developmental markers at this age have been found to be predictive of intelligence, school performance, adult nutrition and human capital in high-income, middle-income and low-income settings[29–31]; this age also corresponds to the end of Piaget's sensorimotor stage.[32] We acknowledge that, while some authors prefer a wider age range for population-based child developmental surveillance,[3] the second birthday remains the earliest time point at which a holistic snapshot of a child's developmental repertoire can be captured reliably and parsimoniously at scale,[15] while still *within* 'the golden window of opportunity for neurodevelopment rescue'—the first 3 years of life—when interventions are evidenced to yield considerable benefits.[33] Conversely, some may argue that the 22–30 months age range is too broad in the context of the rapidly developing nervous system. By evaluating the performance of the INTER-NDA against the BSID-III in children aged 22–30 months,[16] we have provided evidence that the INTER-NDA is a valid and reliable measure of child development in this age group. Nevertheless, the INTER-NDA is a standardised screening assessment and does not provide a clinical diagnosis. Therefore, the possibility for misclassification must be considered when interpreting the findings. A further limitation is that three of the INTERGROWTH-21st Project study sites (China, Oman and the USA) did not participate in the neurodevelopmental evaluation. While the inclusion of these sites might have increased our overall sample size; as evidenced by the WHO MGRS study, the representation of every country is not necessary for the construction of biological standards because of the inherent prescriptive nature of the cohort.[6 34] Our findings, published in 2014 and earlier this year, confirmed that the growth and development of children across different ancestries, geographies and cultures are very similar from early pregnancy to 2 years of age, when environmental constraints on their health and nutrition are minimal, and justified the pooling of data across the five populations for the construction of international standards.[23 24] An additional limitation of our study is the exclusion of the detailed auditory and actigraphy data from the analyses.

To address the question as to what limits should be applied to determine thresholds of normal and non-normal development,[12] we were guided by other neurodevelopmental tools using centile ranks to stratify NDD risk (online supplementary information S1). While many of these define suboptimal development as below the 25th centile, we have presented evidence that most children in our cohort were developmentally normal for age. Therefore, we selected a lower threshold (≥10th centile) to define neurodevelopmental normality. Nevertheless, we acknowledge that, in clinical practice, risk-threshold determination may often take into consideration other factors such as parental concerns and resource allocation.[12]

**Table 5** Evaluation of the INTER-NDA against pre-established feasibility criteria for use of an early child development assessment in a low-income and middle-income setting

| | Does INTER-NDA fulfil the criteria? | Additional details |
|---|---|---|
| **World Bank Toolkit for Examining ECD*** | | |
| Psychometrically adequate, valid and reliable | Yes | ICCs 0.74 and 0.88 (p<0.001) between BSID-III and INTER-NDA for cognitive, language and motor domains; internal consistency 0.56–0.80†. Inter-rater reliability: k = 0.70, 95% CI 0.47 to 0.88); test–retest reliability: k=0.79, 95% CI 0.48 to 0.96‡. |
| Balanced in terms of no of items at the lower end to avoid children with low scores | Yes | Age range of items: 6–36 months‡ |
| Enjoyable for children to take (eg, interactive, colourful materials) | Yes | |
| Relatively easy to adapt to various cultures | Yes | Adapted via cultural customisation session during training and currently in use in 12 countries (Brazil, India, Italy, Kenya, Pakistan, Thailand, South Africa, Mexico, Grenada, Finland, Guatemala, Democratic Republic of Congo; www.inter-nda.com) |
| Easy to use in low-resource settings; for example, not requiring much material | Yes | See Murray et al, for image of kit†; cost <GBP 120.00; no fee per use; manuals and forms freely available at www.intergrowth21.org.uk |
| Not too difficult to obtain or too expensive | Yes | See above |
| Able to be used in a wide age range | Moderately narrow age range | 22–30 months |
| **Fischer et al's feasibility criteria for use of developmental screening tools at primary healthcare level in low-middle income settings§** | | |
| Results understood by health workers | Yes | Centiles |
| Reliable | Yes | See above |
| Valid | Yes | See above |
| Acceptable to caregivers | Yes | |
| Provides information that is relevant to primary care providers | Yes | Centiles |
| Information that can be used for referrals of early intervention | Yes | Centiles |
| Information that is useful for anticipatory guidance | Unknown | |
| Results understood by caregivers | Yes | |
| Staff members have the expertise to answer questions | Yes | Session on maternal questions and responses included in training package. |
| Access to application | Yes | Freely accessible at www.intergrowth-21.org.uk |
| Training involved | Yes | Time taken to train assessors in the INTER-NDA: 1 day for ≤3 assessors, 2 days for 3–5 assessors, 3 days for 5–10 assessors |
| How long it takes to administer the tool | 15 min | |

Continued

**Table 5** Continued

| | Does INTER-NDA fulfil the criteria? | Additional details |
|---|---|---|
| Cover multiple areas of child development | Yes | Cognition, language, fine and gross motor skills and behaviour (positive, negative and global)‡ |
| Cost of the tool | Minimal | Cost of kit <GBP 120.00; no fee per use; manuals and assessment forms freely available at www.intergrowth21.org.uk. Tablet/phone based data collection application (INTER-NDA E-form) optional. |
| Minimal adaptation needed | Yes | Sessions on cultural customisation and translation included in training |
| Educational level of staff members | Secondary education | Results of comparison between field workers and specialists presented in online supplementary table S6 and in text. |
| How many staff members to administer the tool | 1 | |
| Local norms available | International references available | Normative sample drawn from a prospectively recruited sample of 2 years from Brazil, India, Italy, Kenya and the UK with confirmed optimal nutritional, health and developmental status during the first 1000 days of life. |
| Space | Minimal | Storage of kit and forms/table. See Murray et al, for image of kit.† |

*Fernald et al.[3]
†Murray et al.[16]
‡Fernandes et al.[15]
§Fischer et al.[4]

BSID, Bayley Scales of Infant Development; ECD, early child development; ICCs, intraclass correlations; INTER-NDA, INTERGROWTH-21st Neurodevelopment Assessment.

## Context of the study

Measuring neurodevelopmental milestones during early childhood at scale and comparing outcomes across populations are essential prerequisites for achieving the United Nations Sustainable Development Goal (UN SDG) 4.2 ('ensure that all girls and boys have access to quality early child development, care and preprimary education so that they are ready for primary education'). The international INTER-NDA standards presented here contribute an important component to the care of young children: a unique clinical tool for use across all healthcare systems (table 5) to measure neurodevelopmental milestones and associated behaviours in 2-year-old uniformly and at scale, and to identify children at risk of NDDs who would benefit from specialist referral and further investigation (figure 3). It is hoped that these INTER-NDA standards, complementing our published standards for fetal growth and newborn size, and the WHO Child Growth Standards, will (1) contribute to the attainment of the early child development components of the UN SDGs and the WHO survive, thrive, and transform goals of the Global Strategy on Women's, Children's and Adolescents' Health and (2) provide a methodological template for the extension of the construction of child developmental standards to younger and older age groups.

## CONCLUSION

From this international, population-based cohort of healthy and well-nourished children, confirmed to be at low-risk of adverse health, growth and developmental outcomes during the first 2 years of life, we have constructed the first international standards for cognition, language, motor skills and behaviour at 2 years of age measured on the INTER-NDA. The use of standards to measure early child development is superior to references because of their prescriptive nature and universal applicability, in a manner similar to growth standards.

**Author affiliations**
[1]Faculty of Medicine, Department of Paediatrics, University of Southampton, Southampton, UK
[2]Nuffield Department of Women's & Reproductive Health, University of Oxford, Oxford, UK
[3]Oxford Maternal & Perinatal Health Institute, Green Templeton College, University of Oxford, Oxford, UK
[4]Section of Child and Adolescent Psychiatry, Department of Psychiatry, Warneford Hospital, University of Oxford, Oxford, UK
[5]Johns Hopkins Bloomberg School of Public Health, Johns Hopkins University, Baltimore, Maryland, USA
[6]Postgraduate Program in Epidemiology, Federal University of Pelotas, Pelotas, Rio Grande do Sul, Brazil
[7]Post-Graduate Program in Health and Behavior, Universidade Catolica de Pelotas, Pelotas, Rio Grande do Sul, Brazil
[8]Dipartimento di Scienze Pediatriche e dell' Adolescenza, SCDU Neonatologia, Universita di Torino, Torino, Piemonte, Italy
[9]Nagpur INTERGROWTH-21st Research Centre, Ketkar Hospital, Nagpur, India
[10]Faculty of Health Sciences, Aga Khan University, Nairobi, Kenya
[11]Ospedale Infantile Regina Margherita, Sant'Anna Citta della Salute e della Scienza di Torino, Torino, Piemonte, Italy
[12]Department of Radiation Sciences, Umeå University, Umeå, Sweden
[13]Wallenberg Centre for Molecular Medicine (WCMM), Umeå University, Umeå, Sweden
[14]Neurosciences Unit, KEMRI-Wellcome Trust Research Programme, Kilifi, Kenya
[15]Department of Psychology, United States International University, Nairobi, Kenya
[16]College of Health Sciences, University of Sharjah, Sharjah, UAE
[17]Centro de Tecnologia e Innovacion, Mexico City, Mexico
[18]Faculty of Medicine, Universidade Federal de Pelotas, Pelotas, Rio Grande do Sul, Brazil
[19]Centre for Statistics in Medicine, Nuffield Department of Orthopaedics, Rheumatology & Musculoskeletal Sciences, University of Oxford, Oxford, UK
[20]Departments of Obstetrics and Gynecology and of Global Health, University of Washington, Seattle, Washington, USA
[21]School of Public Health, Peking University, Beijing, China
[22]Department of Family & Community Health, Ministry of Health, Muscat, Oman
[23]Department of Engineering Science, University of Oxford, Oxford, UK
[24]Centre for Global Child Health, The Hospital for Sick Children, Toronto, Ontario, Canada

**Acknowledgements** We gratefully acknowledge the Health Authorities in Pelotas, Brazil; Nagpur, India; Turin, Italy; Nairobi, Kenya and Oxford, UK who facilitated this Project by permitting the participation of the respective study sites as collaborating centres. We are grateful to MedSciNet, UK, and Centro de Tecnología e Innovación, Mexico, for their support in the development, customisation and maintenance of the Project database and the NeuroApp, respectively. We are extremely grateful to the participating mothers and children in all the study sites, and to all those who contributed to the development of the INTERGROWTH-21st Project protocol.

**Contributors** JV and SK conceptualised and designed the INTERGROWTH-21st Project. MF, JV, SK, AS, LA, AAA, MK, FG and KW designed and developed the FGLS neurodevelopmental follow-up and the INTERGROWTH-21st Neurodevelopment Assessment (INTER-NDA). JV, SK, CGV, FCB, ZB, CG, EB, ATP, MGG, RP, YAJ, LCI and EB were responsible for the implementation all aspects of the INTERGROWTH-21st Project. MF, JV, SK and LCI coordinated the implementation of the neurodevelopmental follow-up. MF, LCI, FCB, EA, PDN, EB, FG, MP, NK, RO, MC, TS, ST, EM, AL and AW were responsible for site-based training and data collection. JV, SK, ESU, EOO and MF were responsible for data management. MF, JV, ESU, EOO and SK had access to the Project's data; JV, ESU, MF, SK and EOO were responsible for the statistical analysis. MF and JV wrote the report with input from all the other authors. All authors reviewed and approved the final manuscript. SK and JV are responsible for the overall content as guarantors. The guarantors accept full responsibility for the work and/or the conduct of the study, had access to the data, and controlled the decision to publish. The corresponding author attests that all listed authors meet authorship criteria and that no others meeting the criteria have been omitted.

**Funding** This work was supported by a grant from the Bill & Melinda Gates Foundation to the University of Oxford, Oxford, UK (Grant ID# 49038). MF is supported by an Academic Clinical Fellowship in Paediatrics from the National Institute for Health Research, UK to the University of Southampton. ATP is supported by the National Institute for Health Research Oxford Biomedical Research Centre.

**Disclaimer** The funders had no role in the study design, data collection and analysis, decision to publish or preparation of the manuscript.

**Competing interests** None declared.

**Patient consent for publication** Not required.

**Ethics approval** The INTERGROWTH-21st Project was approved by the research ethics committees of the Universidade Federal de Pelotas, Faculdade de Medicina comitê de ética em pesquisa (Ref: OF.051/09), the Indian Ministry of Health and Family Welfare and the Institutional Ethics Committee, Ketkar Hospital, Nagpur (Ref: 5/7/314/2008-RHN); Servizio Sanitario Nazionale – Regione Piemonte, Aziende Ospedaliere OIRM/S.Anna, Oridine Mauriziano di Torino, Comitato Etico Interaziendale (Ref: G9947/CEI/ C.27.2); the Aga Khan University Health Research Ethics Committee, Aga Khan University, Nairobi, Kenya (Ref: AKU- 09-106), and the Oxfordshire Research Ethics Committee 'C', UK (Ref: 08/H0606/139).

**Provenance and peer review** Not commissioned; externally peer reviewed.

**Data availability statement** Data are available on reasonable request. Extra data are available by emailing intergrowth21st@tghn.org.

**ORCID iD**
Michelle Fernandes http://orcid.org/0000-0002-0051-3389

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
