## [Reviewer comments · BMJ Open]

ARTICLE DETAILS

TITLE (PROVISIONAL)	The INTERGROWTH-21st Project International INTER-NDA standards for child development at two years of age: An International Prospective Population-based Study
AUTHORS	Fernandes, Michelle; Villar, José; Stein, Alan; Staines Urias, Eleonora; Garza, Cutberto; Victora, Cesar; Barros, Fernando; Bertino, Enrico; Purwar, Manorama; Carvalho, Maria; Giuliani, Francesca; Wulff, Katharina; Abubakar, Amina A.; Kihara, Michael; Cheikh Ismail, Leila; Aranzeta, Luis; Albernaz, Elaine; Kunnawar, Naina; Di Nicola, Paola; Ochieng, Roseline; Sandells, Tamsin; Savini, Sandy; Temple, Sophie; Murray, Elizabeth; Ohuma, Eric O; Gravett, MG; Pang, Ruyan; Jaffer, Yasmine; Noble, J. Alison; Winsey, Adele; Lambert, Ann; Papageorghiou, Aris; Bhutta, Zulfiqar; Kennedy, Stephen

VERSION 1 – REVIEW

REVIEWER	Pilar Medina-Alva Instituto Materno Perinatal, Lima, Peru
REVIEW RETURNED	17-Dec-2019

GENERAL COMMENTS	Large international cohort and a tool that seems to be easy to use in different settings. Should be extended for 4-7 year olds, maybe with a wider range of international populations.
--

REVIEWER	Tor Strand Innland Hospital Trust Norway
REVIEW RETURNED	30-Dec-2019

GENERAL COMMENTS	The INTERGROWTH-21st Project International INTER-NDA standards for child development at two years of age: An International Prospective Population-based Study This protocol paper describes the construction of the child development standards from 5 locations of the Intergrowth-21st project. This project is a multi centre maternal and child cohort study undertaken in eight countries worldwide. The main purpose of the parental study was to study growth, health, and neurodevelopment up to the age of 2 years. The project has resulted in several highly cited publications. Child development was a predefined outcome from this project, and it makes sense to use this opportunity to constrict age-appropriate standards for the different Scaled neurodevelopmental domains. This effort complements the previously published standards for fetal growth from this project.
--

	The manuscript is wonderfully written and well structured and provides an appropriate discussion on the strengths and limitations of this project and the chosen approach. Specific comments In the results section, some of the decimal separators are placed differently (Lancet style). There are also some other typos in this section and inconsistent use of numbers. Please revisit. Please check names in reference 2 (OrganziationWHO WH), 29 ("Team") Reference 34 lacks year
--	--

REVIEWER	Susan Clifford Murdoch Children's Research Institute, Australia
REVIEW RETURNED	14-Feb-2020

GENERAL COMMENTS	Thanks for the opportunity to comment on this paper. This is a very clear, informative paper presenting developmental standards for healthy 2 year olds, including the detailed methods and rationale for their development. The authors should be commended for conducting the study to a rigorous and high standard. The paper is very well written and a particular strength is the clear rationale provided for methodology decisions. My only minor suggestion for change is to add the total range of the INTER-NDA Protocol Adherence Score to table S6 so readers can interpret the median score. Thank you
--

VERSION 1 – AUTHOR RESPONSE

Reviewer(s) Reports:

Reviewer: 1

Reviewer Name

Pilar Medina-Alva

Institution and Country

Instituto Materno Perinatal, Lima, Peru

Please state any competing interests or state 'None declared':

None declared

Please leave your comments for the authors below

Large international cohort and a tool that seems to be easy to use in different settings. Should be extended for 4-7 year olds, maybe with a wider range of international populations.

RESPONSE:

We thank Dr. Medina-Alva for her comments. We hope to extend this work both to younger and older age groups of children, and to a wider range of international populations.

Reviewer: 2

Reviewer Name

Tor Strand

Institution and Country

Innland Hospital Trust
Norway

Please state any competing interests or state 'None declared':
None declared

Please leave your comments for the authors below

The INTERGROWTH-21st Project International INTER-NDA standards for child development at two years of age: An International Prospective Population-based Study

This protocol paper describes the construction of the child development standards from 5 locations of the Intergrowth-21st project. This project is a multi centre maternal and child cohort study undertaken in eight countries worldwide. The main purpose of the parental study was to study growth, health, and neurodevelopment up to the age of 2 years. The project has resulted in several highly cited publications.

Child development was a predefined outcome from this project, and it makes sense to use this opportunity to constrict age-appropriate standards for the different Scaled neurodevelopmental domains. This effort complements the previously published standards for fetal growth from this project.

The manuscript is wonderfully written and well structured and provides an appropriate discussion on the strengths and limitations of this project and the chosen approach.

Specific comments

In the results section, some of the decimal separators are placed differently (Lancet style). There are also some other typos in this section and inconsistent use of numbers. Please revisit. Please check names in reference 2 (OrganziationWHO WH), 29 ("Team")
Reference 34 lacks year

RESPONSE:

We thank Dr Strand for his positive comments.

We are grateful for his attention to detail and have addressed the decimal separators in all documents related to the submission. We have reviewed the results section carefully, and have addressed typos and inconsistencies.

We have checked the references and amended reference 2 and 34. We also noted that the reference Papageorghiou AT et al, The Lancet. 2014 occurred twice in our reference list, we have amended this error.

Reviewer: 3

Reviewer Name

Susan Clifford

Institution and Country

Murdoch Children's Research Institute, Australia

Please state any competing interests or state 'None declared':

None declared.

Please leave your comments for the authors below

Thanks for the opportunity to comment on this paper. This is a very clear, informative paper presenting developmental standards for healthy 2 year olds, including the detailed methods and rationale for their development. The authors should be commended for conducting the study to a rigorous and high standard. The paper is very well written and a particular strength is the clear rationale provided for methodology decisions. My only minor suggestion for change is to add the total range of the INTER-NDA Protocol Adherence Score to table S6 so readers can interpret the median score. Thank you

RESPONSE:

We thank Dr Clifford for her generous comments. We had added in the total range of the INTER-NDA Protocol Adherence Score to table S6.